# Modeling manual wheelchair propulsion cost during straight and curvilinear trajectories

Jacob Misch[1,2]*, Morris Huang[1,3], Stephen Sprigle[1,2,4]

**1** Rehabilitation Engineering and Applied Research Laboratory, Georgia Institute of Technology, Atlanta, Georgia, United States of America, **2** School of Mechanical Engineering, Georgia Institute of Technology, Atlanta, Georgia, United States of America, **3** Department of Bioengineering, University of Colorado Denver, Denver, Colorado, United States of America, **4** Schools of Mechanical Engineering and Industrial Design, Georgia Institute of Technology, Atlanta, Georgia, United States of America

* mischjp@gatech.edu

**Data Availability Statement:** All relevant data are available at https://hdl.handle.net/1853/62572.

**Funding:** This work was supported by the Rehabilitation Engineering and Applied Research Lab through internal funding.

## Abstract

Minimizing the effort to propel a manual wheelchair is important to all users in order to optimize the efficiency of maneuvering throughout the day. Assessing the propulsion cost of wheelchairs as a mechanical system is a key aspect of understanding the influences of wheelchair design and configuration. The objective of this study was to model the relationships between inertial and energy-loss parameters to the mechanical propulsion cost across different wheelchair configurations during straight and curvilinear trajectories. Inertial parameters of an occupied wheelchair and energy loss parameters of drive wheels and casters were entered into regression models representing three different maneuvers. A wheelchair-propelling robot was used to measure propulsion cost. General linear models showed strong relationships ($R^2 > 0.84$) between the system-level costs of propulsion and the selected predictor variables representing sources of energy loss and inertial influences. System energy loss parameters were significant predictors in all three maneuvers. Yaw inertia was also a significant predictor during zero-radius turns. The results indicate that simple energy loss measurements can predict system-level performance, and inertial influences are mostly overshadowed by the increased resistive losses caused by added mass, though weight distribution can mitigate some of this added cost.

## Introduction

A substantial body of research has developed from the need to improve manual wheelchair mobility, with the goals of reducing the risk of upper extremity injuries [1–3] and facilitating increased independence and community participation [4]. A primary metric of wheelchair mobility is termed 'effort', which reflects the biomechanical exertion of the human user whilst propelling the wheelchair. Studies on the human propulsion effort, typically measured as metabolic cost, have investigated the impact of different wheelchair types and configurations but fall short of being able to inform clinical decision-making by modifying the tested wheelchairs beyond their standard configuration [5], or using human subject methods that lacked sensitivity [6, 7] to the different wheelchair configurations. A large part of this knowledge gap stems

**Competing interests:** The authors have declared that no competing interests exist.

from a lack of understanding in the relative contributions of the human user's biomechanical effort and the wheelchair's mechanical propulsion cost to the resulting motion. Given the overwhelming diversity in human users and their corresponding propulsion biomechanics in different wheelchairs, the most efficient path is to first focus on developing standardized and representative measurements of the wheelchairs' mechanical propulsion cost. Isolating the wheelchair performance from the inherently high-loss human biomechanics offers a standardized measurement of the mechanical system's maximum efficiency capabilities, which is valuable information for clinicians and users during wheelchair selection, and informs researchers how propulsion effort can be impacted by the overall mechanical design of the wheelchair.

Research on manual wheelchair (MWC) mechanical performance has taken the forms of both component- and system-level testing with each type having strengths and weaknesses. Component-level testing is typically comprised of simple methodologies that quantify the resistive forces or losses associated with wheelchair drive wheels and/or casters, key components that are widely considered the largest source of energy loss as described by Schuring [8], Pacejka [9], and Hofstad [10], among others [11, 12]. Researchers have pointedly studied frictional energy loss of the wheels to identify areas of improvement for MWC configuration and design. The rolling resistances of drive wheels have been directly measured using coast-down tests [13–15], treadmills [13, 16], and rollers [17]. Furthermore, Frank and Abel studied the impacts of caster size and shape on the caster rolling resistance [13]. In the same study, they also created a rotational scrub testing rib to investigate the turning resistance torque of each caster. Gordon, Kauzlarich, and Thacker have also studied characteristics of caster and drive wheels including rolling resistance, static friction, and spring rate [16, 18, 19], which built a strong empirical foundation for the significance of component-specific test procedures. Other methods include coast-down tests to study wheel styles, wheelchair mass and mass distribution, and tire inflation pressures [20, 21], as well as measuring swiveling resistance of drive wheels using a varying-radius load arm [22]. More recently, Silva has developed a benchtop methodology to characterize drive wheel corning forces that occur during combined rolling and turning, with various slip and camber angles [23]. Component-level tests are repeatable and offer a valid means to compare different components. However, they often cannot capture the complex interactions that take place at the system-level during over-ground maneuvering, which limits their direct clinical relevance.

Systems-level tests of MWC propulsion characteristics and energy losses expand on component testing to permit assessment of various wheelchair configurations in more realistic usage conditions. These include tests both with and without human operators. Systems-level testing with human operators has been dominated by focus on steady-state velocity often using treadmills and rollers [3, 7, 24–27], which are not well positioned to assess the propulsion effort required to maneuver a wheelchair over-ground. Tests using human operators during over-ground maneuvers offer a much more realistic representation of propulsion forces and effort, as the system is endowed with changes in momentum and travels over common surface types. These studies have assessed the impact of different surfaces [5, 28, 29], wheelchair mass [5, 6], weight distribution or axle position [5, 30, 31], and push forces at different speeds [32, 33]. Systems-level research without human operators affords a unique opportunity to investigate the mechanical behavior and performance of the vehicle with repeatable, reliable, and systematic methodologies. In 2004, Sawatzky used an inclined coast-down test to observe the impacts of various drive wheel styles and inflation pressures on the system-level rolling resistance of a lightweight wheelchair [20]. Similarly, to determine the system resistance to motion, Van der Woude measured the magnitude of external force applied to handlebars behind the seat that was required to roll wheelchairs at different constant speeds across multiple surfaces [29]. Systems-level testing, to date, has often been constrained to straight and steady-state conditions

that lack representation in real-world wheelchair mobility [34], though there are a few notable exceptions. For example, Sauret and Bascou measured the field rolling resistance of loaded MWCs in very similar fashion to component test protocols [14], and by utilizing the actual MWC instead of a cart or test rig, they had the ability to expand into curvilinear coast-down tests and assess the turning resistances of various configurations [35]. Other examples include Lin's curvilinear coast-down testing [21] and Sprigle and Huang's wheelchair-propelling robot that was used to identify differences in propulsion torque to accelerate different MWCs [36]. These cited systems-level methodologies measure under discrete and curvilinear maneuver conditions but often utilize complex, customized testbeds that are not easily reproducible, such as in [36]. However, the benefits of using this particular wheelchair-propelling robot instead of human subjects are numerous: the system parameters of mass and weight distribution can be controlled; the MWC, as a vehicle, can be studied independent of the biomechanics of the human operator; many more configurations can be studied; the instrumented system has demonstrated high test repeatability [37] as well as sensitivity to MWC configuration changes [36, 38].

The limiting factor of most systems-level test methods is that it becomes more difficult to identify the most significant factors comprising differences in mechanical performance. Studies on wheelchair modeling have attempted to bridge this knowledge gap between component-specific results and systems-level test results. Most cases have involved applying measured component-level parameters in a theoretical wheelchair model to simulate some aspect of the vehicular dynamics, such as the use of turning and rolling resistance values from [23] in a computational model of the lateral forces acting on a moving wheelchair [39]. Caspall et al. used inertial and geometric measurements of MWCs to model the turning resistance during zero-radius turns with a human operator [40]. Sauret, Bascou, and Fallot have generated mathematical models using the rolling resistance [14] and turning resistance [22, 35] measurements to assess energy losses through non-conservative ground contact forces [41]. Medola et al. developed a mathematical model to partition MWC kinetic energy into translational, rotational, and turning energies using geometric measurements and inertial parameters of system components [42].

Many computational wheelchair models have been created to directly relate easier-to-measure component test results to system-level propulsion cost [39, 40, 42–44]. As in [10], these models serve to isolate the variables that best predict the efficiencies of each chair, and allow readers to infer the relative propulsive forces or effort required to move the MWCs. In many ways, this work mirrors the use of dynamic models in the automotive [9, 12] and cycling [45–47] industries to inform the design of the chassis, frame, suspension, and optimize stability. One early wheelchair model of straight trajectory racing wheelchair propulsion attempted to calculate dynamic system output using component metrics, biomechanics, chair geometry, and aerodynamics to the dynamic system output [48]. This model was later evaluated by Hofstad and Patterson who concluded that no single model could accurately predict forces during both rectilinear and curvilinear motion, though certain parameters such as bearing resistance and air drag were not significant predictive factors in either type of motion [10].

Efforts to improve the understanding of MWC mechanics during curvilinear motion have been conducted by observing the contact forces at the interface between the wheel and a length of test surface with various slip and camber angles [23]. This team later used these turning and rolling resistance values into a computational model of the lateral forces of simple wheelchair motion [39], though the test speed, load, and surface choices were not explained. Chénier et al. generated a method to estimate caster orientation from drive wheel measurements [49], and improved upon the general linear model in predicting wheelchair turning motion [44]. However, their applied model of curvilinear motion placed all of the system rolling resistance on the front casters, assuming drive wheel contributions were negligible. More recent attempts by

Bascou postulated that MWC system resistance could be broken into contributions by each wheel and that these contributions could be further divided into rolling and turning resistances [35, 43]. Given the complexity of wheelchair dynamics and the limitations of theoretical models, there exists a need to develop and validate models of propulsion cost with straight and curvilinear trajectories that use component- and system-level 'predictors'.

The reviewed works above suggest that a need exists for manual wheelchair modeling that combines the ease and adaptability of bench testing of components and system parameters with the in-situ context of a repeatable system-level test, such as the robotic wheelchair propulsion system described in [36–38]. Test methods independent of human operators remove any influence of human biomechanics and instead focus on the vehicular dynamics to differentiate the impacts of inertial and energy loss parameters on MWC propulsion. Ultimately, the goal of this study was to model the relationships between inertial and energy loss parameters to the mechanical propulsion cost. These 'predictor' variables, composed of component- and system-level parameters, include rolling resistance, tire scrub, system mass, yaw inertia, and weight distribution across different wheelchair configurations and maneuvers. The resulting relationships identified by the statistical modeling effort will create an opportunity for manufacturers and researchers to firstly identify relative contributions of MWC configuration choices, and secondly to better estimate MWC performance using simple coast-down and scrub torque tests. The increased accessibility of this information will promote more educated prescription and design of wheelchairs to fit the broad range of wheelchair users.

## Background and prior works

Measurement of bouts of mobility in a MWC during everyday life have found to be relatively are short (median = 20 secs) and slow (median = 0.44 m/s) with frequent stops and turns [34]. This informs the testing of energy loss and propulsion cost by indicating the need to include changes in momentum (speed and direction). Medola partitioned the kinetic energy of a human-driven MWC over flat surfaces to show how translational, rotational, and yaw (turning) energies interchanged between MWC trajectories, including a straight trajectory, a slalom and 2-m radius turn, to highlight the differences between MWC maneuvers [42]. Likewise, Huang studied three paramount types of motion as straight-line, fixed-radius turning, and zero-radius turning. By the tenets of physics, these 'canonical maneuvers' distinctly isolate the impacts of inertia and energy loss, specifically rectilinear and yaw inertia, rolling resistance and scrub-resistive torque [50].

## Overarching approach

This work describes the use of inertial and energy loss parameters, measured at the component- and system-level, as predictors for a statistical model of empirically-measured MWC mechanical propulsion cost. The design of the overarching modeling approach is based on the dynamical models of three distinct canonical maneuvers, selected to collectively embody the inertias and energy losses inherent in everyday wheelchair mobility. These canonical maneuvers are illustrated below (**Fig 1**).

These maneuvers are not necessarily the motions that are individually performed regularly during everyday mobility. Rather, everyday MWC motion consists of bouts of combinations of straight motion and turning with various radii of curvature. However, the maneuvers above were defined to represent the general and most common inertial and energy loss parameters that dictate the effort to propel manual wheelchairs. The canonical maneuvers could then be used, in combination, to represent more complex common everyday maneuvers. To achieve a simplified, but representative, cross-section of the diverse types of MWC motion, the

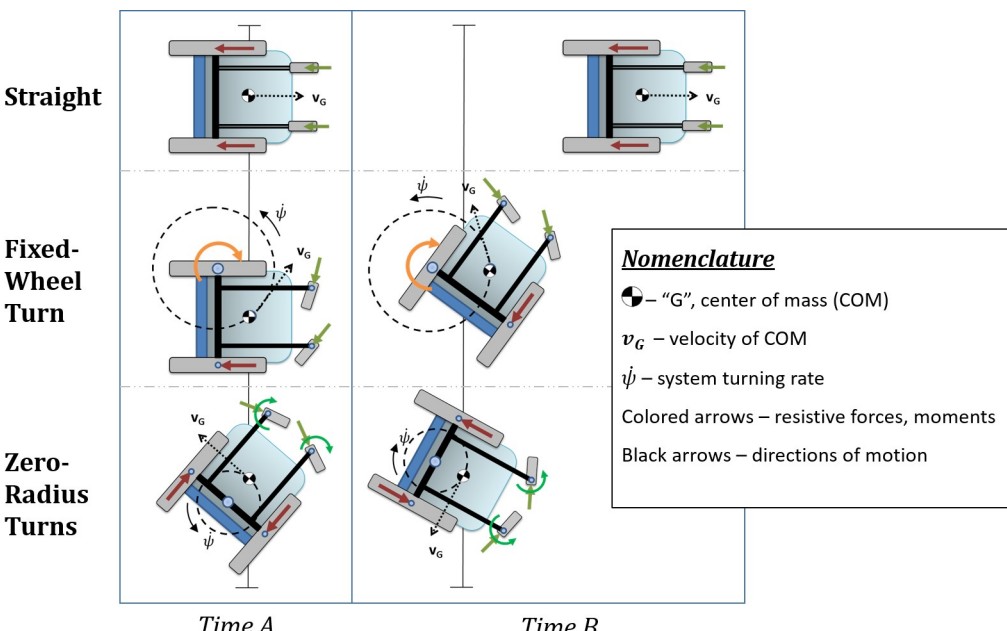

**Fig 1. Canonical maneuvers.** Graphical depictions of the 'canonical maneuvers' used to embody the most common features of MWC motion.

maneuvers include a straight trajectory and two turning trajectories embodied by different radii of curvature. In pure straight-line motion, rectilinear inertia and rolling resistance are the most relevant parameters to the dynamics of the system, whereas scrub torque and yaw inertia will offer negligible contributions. Similarly, when performing a fixed-wheel turn, the casters will align and roll along the curvature of motion with negligible scrub, the rolling drive wheel will experience minimal scrub, and the fixed drive wheel will exclusively experience scrub. Excluding other non-conservative forces or torques that serve as sources of energy loss, the knowledge of rolling resistance and scrub torque should suffice as an adequate predictor of the system energy loss, and therefore the cost of travel for any given MWC may be estimated through simple component-level bench testing. These non-conservative forces and torques are illustrated below (**Fig 2**).

**Fig 1** and **Fig 2** illustrate how each component resistance, when coupled with wheelchair dimension measurements, contributes to the overall system resistance of each modeled maneuver (exact mathematical relationships are defined in "Modeling", below). Note that some resistive force terms are omitted from these diagrams, such as the resistive scrub torques for the drive wheels as they are rolling and turning in the fixed-wheel turn and zero-radius turn maneuvers. The rationale for these omissions is that these terms involving combined rolling and turning are not reflected in the component tests (described in the "Component-level testing" section), and more importantly, are believed to negligible relative to the included resistive force terms when applied towards our modeling purposes. The quality of the regression models supports this decision in simplifying the definitions for system resistance.

## Methods

To populate the models with predictor variables, four separate testing protocols were utilized to measure rolling resistance force, turning scrub resistive torque, system-level MWC propulsion cost, and inertial parameters.

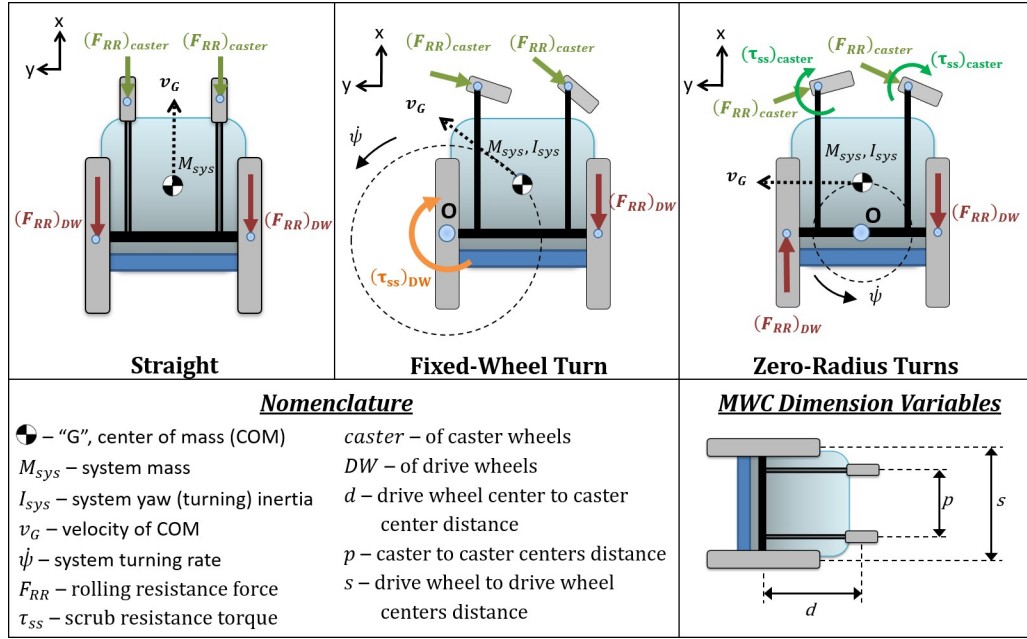

**Fig 2. Simplified MWC diagrams.** Simplified diagrams displaying the most significant non-conservative forces and torques acting at the interfaces between the ground and the casters and drive wheels.

## Component-level testing: Coast-down cart

Rolling resistance force ($F_{RR}$) measurements were taken using an instrumented coast-down cart [15] designed to accommodate both 24"-dia drive wheels and 4"-, 5"-, and 6"-dia caster wheels, seen in **Fig 3**. The cart is rapidly accelerated from rest until it reaches a release velocity of around 1.1 m/s. The operator then releases the cart as it freely coasts to a stop. The deceleration of the cart is calculated by finding the slope of the velocity curve within the window of velocities between 0.95 m/s and 0.65 m/s. This window was chosen because the cart consistently has a linear velocity slope within this region. 10 trials are conducted in each direction along the floor and averaged to account for any surface irregularities such as floor slope or bumps.

The cart was loaded with weights to represent realistic weight applications on the 'reference' and 'test' wheels (see **Table 1**, below). Rolling resistance force on each 'reference' wheel, $(F_{RR})_{ref}$, was first calculated by equipping the cart with four identical reference wheels. The total cart mass and the measured average deceleration values are divided by the four identical reference wheels in **Eq 1** to determine the rolling resistance on a single reference wheel.

$$(F_{RR})_{ref} = \frac{Ma_{cart}}{4} \tag{1}$$

Next, to determine RR force on a 'test' wheel, $(F_{RR})_{test}$, the two rear reference caster wheels were replaced by a set of test caster or drive wheels. The deceleration data from the coast-down trials are used in **Eq 2**. The known RR force from the reference wheels is subtracted out. The remaining RR force is divided evenly between the two test wheels.

$$(F_{RR})_{test} = \frac{Ma_{cart} - 2(F_{RR})_{ref}}{2} \tag{2}$$

A more complete description of the coast-down protocol can be found in [15].

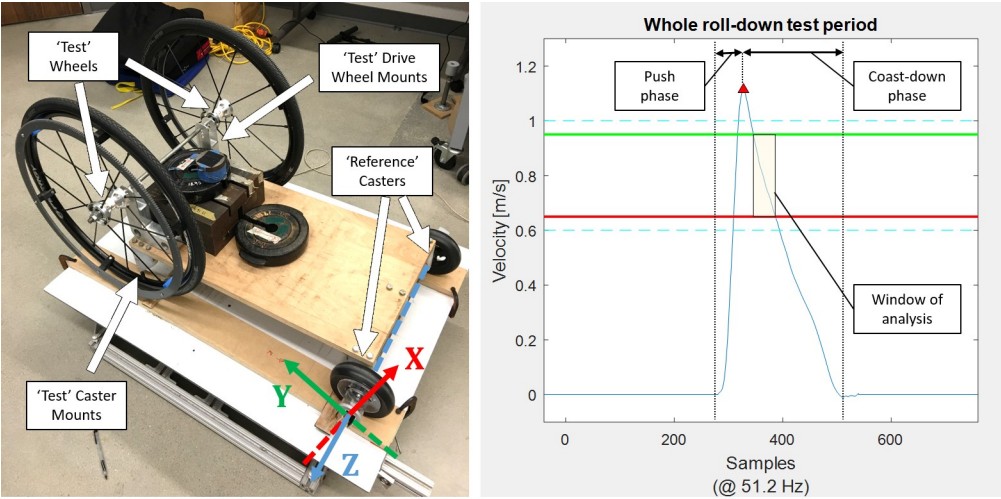

**Fig 3. Coast-down cart.** (Left) Standard loading on coast-down cart for testing drive wheels. The front casters are the 'reference' wheels with equipped accelerometer mounts. (Right) Representative linear velocity of the cart during a single trial. The window of analysis to determine the deceleration value is between 0.95 m/s and 0.65 m/s, denoted by the yellow shaded box.

## Component-level testing: Resistive scrub torque rig

Scrub torque ($\tau_{ss}$) was measured using a materials-testing load frame with a scrub test plate. A custom rig was made to hold 4"-, 5"-, and 6"-dia casters or 24"-dia drive wheels stationary. A small section of the test surface (linoleum tile, low-pile carpet) was attached to a scrub plate and bearing-mounted to a weighted load arm. A length of steel cable attached to the crosshead of the materials tester loops about a pulley and connects to a spool on the scrub plate; as the crosshead rises, the cable becomes taut and causes rotation of the scrub plate against the top surface of the captive wheel. Force of the pull is recorded via the built-in load cell on the cross-head and multiplied by the radius of the spool to determine the representative $\tau_{ss}$ value. In accordance with ASTM Standards E1337-90 A and D1349-14 A, the test surface of the wheel is first cleaned and the wheel is conditioned by performing a single scrub trial to remove the sheen from the rubber [51, 52]. This test protocol has been described previously in greater detail in [15].

## System-level testing of propulsion cost

A wheelchair-propelling robot, denoted as the Anatomical Model Propulsion System (AMPS), was used to propel the test wheelchair. The robotic system removes confounding inherent

**Table 1. Description of system masses and weight distributions.** The values in these columns represent the typical loading experienced by each component during coast-down and scrub torque tests.

| Simulated System Mass | Percent Load on Drive Wheels | Normal Load on Each Caster | Normal Load on Each Drive Wheel |
|---|---|---|---|
| 100 kg | 60% | 20 kg | 30 kg |
| | 70% | 15 kg | 35 kg |
| | 80% | 10 kg | 40 kg |
| 80 kg | 60% | 16 kg | 24 kg |
| | 70% | 12 kg | 28 kg |
| | 80% | 8 kg | 32 kg |

errors associated with human biomechanics to repeatability isolate and assess mechanical efficiencies of MWC configurations and designs. The skeletal anthropometric structure follows the weight distributions according to average body segment parameters and ISO 7176–11 [53] to replicate realistic loads on the wheels, frame, and bearings. The only necessary modifications to interface AMPS with a chair are to replace the hand rim with a custom-made ring gear and affix the hub-mounted drive wheel encoders (M-260 Accu-Coder). One DC motor on each side meshes with the ring gear. The controlling laptop runs a custom LabVIEW (National Instruments) program to generate the propulsion trajectory and sends commands to the motor controller (HDC2450, RoboTeq Inc.) operating in closed-loop speed control. AMPS is instrumented with current sensors (ACS758xCB) and encoders to measure motor torque and drive wheel velocity during the maneuver. An image of the AMPS is provided in **Fig 4** for reference.

A full description of the design and validation of the AMPS was previously reported by Liles to investigate propulsive work [37, 54]. In the current study, this propulsive work term is normalized by the distance traveled over the maneuver to determine the propulsive cost in Joules of energy per meter (J/m).

## Inertial parameters

The total mass of the entire wheelchair system while occupied by the AMPS was measured by an inertial measurement device known as the iMachine [55]. The device consists of a spring-loaded turntable mounted to a single axle. A slight displacement induced a damped harmonic response whose frequency was captured to calculate the rotational (yaw) inertia. Load cells mounted on the turntable measure the mass and the location of the center of mass. The weight distribution of the wheelchair system (%DW) was represented as the percent (%) loading on the drive wheels. The iMachine measured the location of the center of mass (CoM) and %DW loading was then calculated based on the wheelbase.

Drive wheel and caster inertias were quantified with a ceiling-mounted trifilar pendulum, as in [56]. The mass of a single wheel was first measured by a scale. The wheel was then centered on the platform and fastened by aligning a vertical bolt through the axle bearing. A digital camera was placed underneath the hanging platform to record the damped harmonic response of the platform following a gentle perturbation to the angular position. The angular rotations were used to calculate the rotational inertia of the tested wheel.

## Configurations and components

A wide variety of casters and drive wheels were selected to form a representative cross-section of components used in manual wheelchairs. The selections are graphically shown in **Fig 5**, and included one solid and two pneumatic drive wheel tires and four casters of varying diameters and widths.

Rolling resistance and scrub torque parameters of drive wheels and casters were tested with the protocols described above under three load conditions. The loading conditions were chosen based on clinical relevance and related literature [20, 57, 58] to represent a common and realistic range of weight distributions of a loaded chair, e.g. a 100 kg user with 60% weight distribution means 60 kg is over the drive wheel axle, so each drive wheel is under 30 kg load and each caster is under 20 kg. To ensure the protocols reflected these realistic loads, the components were placed under loads as seen in **Table 1**, below. During resistive scrub torque tests, the values from the 'Normal Load' columns were applied to the components. Similarly for the rolling resistance tests, the coast-down cart required different loading if the 'test' wheels were casters or drive wheels. To simulate a 100 kg system mass with 70% weight distribution, for example, the coast-down cart would require the two 'reference' casters and the two 'test' casters

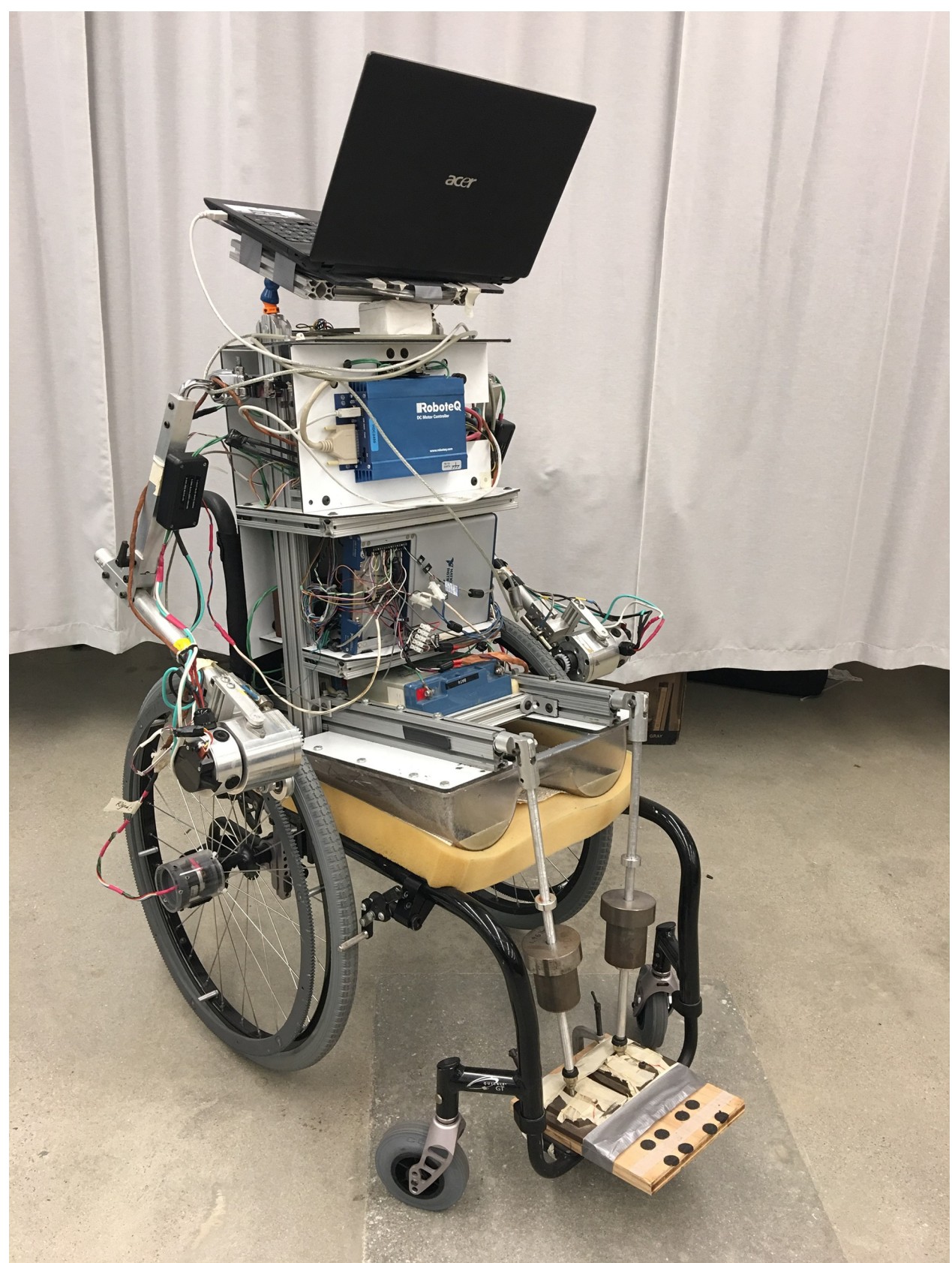

**Fig 4. The AMPS.** The Anatomical Model Propulsion System configured to an ultra-lightweight manual wheelchair.

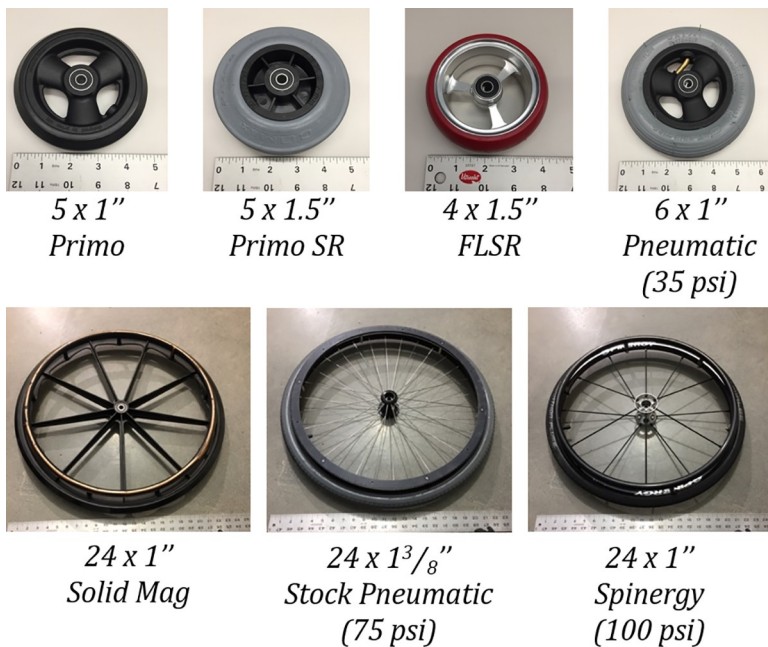

**Fig 5. Tested components.** Array of components representing a range from standard, off-the-shelf wheelchair configurations to costlier up-charged versions.

to be under equal 15 kg loads, so the cart would have a 60 kg mass equally distributed over the four casters. To test the drive wheels, the cart would instead be loaded to 100 kg with 70% of the weight over the drive wheels, such that each drive wheel was under 35 kg load and each of the 'reference' casters was under 15 kg load. The experimental setups are described in greater detail in [15].

## Outcome variable (propulsion cost)

Propulsion cost was selected as the performance metric or outcome variable of the system-level tests. Similar to cost of transport [59, 60], propulsion cost quantifies the amount of energy required to perform a maneuver, normalized by the displacement or distance traveled by the center of mass. First, the propulsion power is calculated by multiplying the left and right wheel propulsion torques ($\tau_L$, $\tau_R$) with the corresponding angular wheel velocities ($\omega_L$, $\omega_R$), measured in real-time by the motor current sensors and wheel-mounted optical encoders, respectively. Then, the AMPS propulsion energy is found by integrating the AMPS power from both sides over the duration of the propulsion task, from an initial time ($t_i$) to a final time ($t_f$). Finally, propulsion cost is found by normalizing the input energy by the displacement of the center of mass ($\Delta s$), as seen in **Eq 3**.

$$Propulsion\ Cost = \frac{\int_{t_i}^{t_f}[(\tau_L * \omega_L) + (\tau_R * \omega_R)]dt}{\Delta s} \tag{3}$$

For the straight maneuvers, $\Delta s$ is the path length traversed by the combined AMPS and wheelchair system center of mass. For fixed-wheel and zero-radius turns, the $\Delta s$ term is replaced with $\Delta \psi$ to represent the total yaw angle traveled, in radians.

Within this study, AMPS trials of both straight and fixed-wheel turn maneuvers used the propulsion cost across the acceleration and steady-state phases for analysis. The acceleration

**Table 2. Outcomes and predictors.** Descriptions of the output variables and the predictor variables for each maneuver.

| Maneuver | Output variable | Inertial predictors | Energy loss predictor |
|---|---|---|---|
| Straight | Main cost; Acceleration cost | Mass, Mass distribution | System loss value (caster and drive wheel rolling resistances) |
| Fixed-Wheel Turn | Main cost; Acceleration cost | Mass, Mass distribution, Yaw inertia | System loss value (caster and drive wheel rolling resistances; drive wheel scrub) |
| Zero-Radius Turn | Total cost | Mass, Mass distribution, Yaw inertia | System loss value (caster and drive wheel rolling resistances; caster scrub) |

phase is defined as the first 2.5 seconds of motion as the AMPS accelerates from rest to the target speed. The next 5 seconds is known as the steady-state phase as the AMPS maintains a constant speed. The "deceleration" or "braking" phase was intentionally omitted from the propulsion cost analysis for the straight and fixed-wheel turn maneuvers, as the goal is to highlight energy expended for task propulsion, not braking. The combination of the acceleration and steady-state phases is labeled as "**Main Cost**" in the Results section, whereas "**Accel Cost**" refers to just the propulsion cost of the acceleration phase. For the zero-radius turns maneuver, the propulsion cost across the last four of the six total turns in each trial was used for analysis, and is referred to as "**Total Cost**". The first two turns were excluded from the analysis window due to their propulsion instabilities from transiently misaligned casters. The propulsion cost of each turn's braking phase is included out of necessity, because zero-radius turns do not exhibit distinct acceleration, steady-state and braking phases. The plotted wheel kinematics of each maneuver type can be found in the supplemental material, **S1 Appendix**.

In total, five models were configured for analysis: separate straight and fixed-wheel turn models were based upon both Main cost and Acceleration cost, and a single model for zero-radius turns was configured for Total cost. The outcome parameter and the most significant predictor variables are shown below in **Table 2**. The predictor variables are explained in greater detail in the following section.

## Predictor variables

To assess the relative influences of inertial and energy loss parameters on the propulsion cost during each maneuver, predictor variables included inertial measurements of the occupied wheelchair and energy loss parameters from the component-level drive wheel and caster testing.

**Configurations.**   The inertial parameters used in the model were comprised of rotational inertias of the drive wheels and casters, measured by trifilar pendulum as in [56], as well as the MWC yaw moment of inertia measurement from the iMachine [55]. Several other parameters were used as categorical variables: the weight of the AMPS, controlled at 80 kg and 100 kg; the weight distributions of 60%, 70%, and 80% of total mass over the drive wheels; the separate caster and drive wheel selections; the test surface of either linoleum tile or low-pile carpet laid flat over polished concrete without underlay padding.

**System loss values.**   A parameter of total system resistance, referred to as System Loss or SysLoss below, was calculated for each trajectory that combined the non-conservative forces and torques from the drive wheels and casters. System Loss is calculated in units of N for straight and N-m for the turning maneuvers to represent the resistive forces acting on the MWC, which are sources of energetic loss in the system. The individual contributions were based upon bench tests of rolling resistance and scrub. The following equations were derived from kinematic [42] and free-body diagrams seen in [61], which are also shown above in **Fig 2**.

For the System Loss term representing energy losses in the straight maneuver, the caster and drive wheel rolling resistance forces denoted by $(F_{RR})_{caster}$ and $(F_{RR})_{DW}$ respectively were combined using the equation:

$$SysLoss_{straight} = 2 * (F_{RR})_{caster} + 2 * (F_{RR})_{DW} \qquad (4)$$

For fixed-wheel turns, drive wheel scrub $(\tau_{ss})_{DW}$ was added to the rolling resistances. Because casters are aligned with the direction of travel, they do not swivel or scrub during the fixed-wheel turn.

$$SysLoss_{FW\ turn} = (\tau_{ss})_{DW} + s * (F_{RR})_{DW} + \left( (F_{RR})_{caster} * \sqrt{\left(\frac{s-p}{2}\right)^2 + d^2} \right)$$
$$+ \left( (F_{RR})_{caster} * \sqrt{\left(\frac{s+p}{2}\right)^2 + d^2} \right) (5)$$

In Eq 5, $s$ represents the width of the wheelbase (distance between drive wheel contact patches), $p$ is the distance between the centers of the caster forks, and $d$ is the fore-aft distance between the center of the drive wheel axle and the center of the caster fork.

For zero-radius turns, caster and drive wheel rolling resistances and caster scrub torques, $(\tau_{ss})_{caster}$, were entered into the calculation. The casters undergo swiveling during each change in direction during a zero-radius turn, and are rolling for the remaining duration of the maneuver. The drive wheels primarily roll and despite how the orientations change, the drive wheel scrub torques are considered negligible in comparison to the other sources of energy loss. Therefore, the drive wheel scrub term was omitted from the equation. The system loss equation for the zero-radius turn maneuver is:

$$SysLoss_{AZR\ turn} = 2 * (\tau_{ss})_{caster} + 2 * (F_{RR})_{DW} * \frac{s}{2} + 2 * \left( (F_{RR})_{caster} * \sqrt{\left(\frac{p}{2}\right)^2 + d^2} \right) \qquad (6)$$

## Modeling and analysis

General Linear Models were run using Minitab 18 to identify predictors of cost of propulsion. System Mass (rectilinear inertia) and Weight Distribution were included in models of all three maneuvers. Yaw Inertia was included in the fixed-wheel turn and zero-radius turn models. The related System Loss parameter and system Yaw Inertia values were entered as continuous covariates into the model for each specific maneuver. Weight Distribution (60%, 70%, 80%) and AMPS mass (80 kg, 100 kg) were entered as categorical variables with 60% WD and 80 kg used as the reference categories. Within each model, coefficients of the continuous predictors were standardized.

Overall, five models were calculated. These models included the propulsion costs during the three canonical maneuvers, as well as additional models of the acceleration phases during the straight and fixed-wheel turning maneuvers, as seen in Table 2. Additional analyses were performed to more fully investigate the various influences of the wheelchair configurations by calculating magnitudes of differences between parameters. The changes in propulsion cost relative to mass and weight distribution were evaluated using effect sizes and graphically depicted using scatter plots and box plots. Effect sizes, reported as Cohen's $d$, were calculated using the difference in the mean values divided by the pooled standard deviation and reflect the magnitude of difference between two groups. Based upon Cohen's general guidelines in [62], effect sizes were classified as small (~0.2), medium (~0.5), large (~0.8) and very large (~1.0).

Table 3. Main cost: Straight trajectory.

**Analysis of Variance**

| Source | DF | Adj SS | Adj MS | F-Value | P-Value |
|---|---|---|---|---|---|
| SYS_LOSS_VALUE | 1 | 7755.3 | 7755.29 | 3304.32 | 0 |
| Error | 142 | 333.3 | 2.35 | - | - |
| Total | 143 | 8088.6 | - | - | - |

**Model Summary with coded coefficients**

| Term | Coef | SE Coef | P-Value | VIF | R-sq(adj) |
|---|---|---|---|---|---|
| Constant | 29.365 | 0.128 | 0 | - | 95.9% |
| SYS_LOSS_VALUE | 7.364 | 0.128 | 0 | 1 | - |

**Regression Equation in Uncoded Units**

MAIN_COST = 14.999 + 1.0280 SYS_LOSS_VALUE

# Results

## Main cost: Straight trajectory

Three parameters were entered into the general linear model analysis: System Loss as a covariate, and Weight Distribution and Mass as categorical factors. The initial results identified System Loss as the sole predictor, so a subsequent model was run using just the System Loss parameter as a covariate (**Table 3**). The results indicate a very strong model with adjusted $R^2$ of 96%.

## Main cost: Fixed-wheel turn

Four predictors of Main Cost were entered into the analysis: System Loss and Yaw Inertia values as continuous covariates, and Weight Distribution and Mass as categorical factors. The results of the initial model identified System Loss as the sole predictor. A subsequent model was run using System Loss alone (**Table 4**) with the resulting model having a $R^2$ of 89%.

## Total cost: Zero-radius turns

Four predictors of Main Cost were entered into the analysis: System Loss and Yaw Inertia values as continuous covariates, and Weight Distribution and Mass as categorical factors. Based upon the results of that model, only System Loss and Yaw Inertia were entered into a subsequent model (**Table 5**) with a resulting $R^2$ of 87%. The coefficients indicate that propulsion cost rises with increases in Yaw Inertia and System Loss, with energy loss being the most influential parameter.

Table 4. Main cost: Fixed-wheel turn.

**Analysis of Variance**

| Source | DF | Adj SS | Adj MS | F-Value | P-Value |
|---|---|---|---|---|---|
| SYS_LOSS_VALUE | 1 | 2404.17 | 2404.17 | 1131.8 | 0 |
| Error | 142 | 301.64 | 2.12 | - | - |
| Total | 143 | 2705.8 | - | - | - |

**Model Summary with coded coefficients**

| Term | Coef | SE Coef | P-Value | VIF | R-sq(adj) |
|---|---|---|---|---|---|
| Constant | 16.453 | 0.121 | 0 | - | 88.77% |
| SYS_LOSS_VALUE | 4.1 | 0.122 | 0 | 1 | - |

**Regression Equation in Uncoded Units**

MAIN_COST = 2.955 + 1.15 SYS_LOSS_VALUE

**Table 5. Total cost: Zero-radius turns.**

| Analysis of Variance | | | | | |
|---|---|---|---|---|---|
| Source | DF | Adj SS | Adj MS | F-Value | P-Value |
| SYS_YAW_INERTIA | 1 | 22.76 | 22.761 | 23.75 | 0.000 |
| SYS_LOSS_VALUE | 1 | 524.21 | 524.206 | 547.08 | 0.000 |
| Error | 141 | 135.11 | 0.958 | - | - |
| Total | 143 | 1056.5 | - | - | - |
| **Model Summary with coded coefficients** | | | | | |
| Term | Coef | SE Coef | P-Value | VIF | R-sq(adj) |
| Constant | 10.994 | 0.0816 | 0 | - | 87.03% |
| SYS_YAW_INERTIA | 0.4704 | 0.0965 | 0 | 1.39 | - |
| SYS_LOSS_VALUE | 2.2575 | 0.0965 | 0 | 1.39 | - |
| **Regression Equation in Uncoded Units** | | | | | |
| TOTAL_COST = 3.281 + 0.3643 SYS_YAW_INERTIA + 0.6515 SYS_LOSS_VALUE | | | | | |

## Acceleration cost: Straight trajectory

Modeling acceleration cost during the straight trajectory entered in System Loss, Weight Distribution and Mass and identified System Loss and Mass as predictors. A subsequent model was run with these two parameters (**Table 6**) resulting in a model with $R^2$ = 94%. The coded coefficients indicate that System Loss is about twice as influential on acceleration cost compared to mass. The resulting equations indicate that a unit increase in System Loss force increases acceleration cost (in J/m) by 27% and an increase in AMPS mass from 80 kg to 100 kg increases cost by 4.7 J/m over all the configurations.

## Acceleration cost: Fixed-wheel turn

For fixed-wheel turns, the acceleration cost models entered System Loss, Yaw inertia, Weight Distribution and Mass. The results of the initial model identified System Loss and Weight Distribution as predictors. The subsequent model run using these predictors (**Table 7**) yielded an output equation with $R^2$ = 85%. The coefficients reflect the importance of energy loss as a

**Table 6. Accel cost: Straight trajectory.**

| Analysis of Variance | | | | | |
|---|---|---|---|---|---|
| Source | DF | Adj SS | Adj MS | F-Value | P-Value |
| SYS_LOSS_VALUE | 1 | 10558.7 | 10558.7 | 1677.87 | 0 |
| Mass | 1 | 720.8 | 720.8 | 114.54 | 0 |
| Error | 141 | 887.3 | 6.3 | - | - |
| Total | 143 | 15694.5 | - | - | - |
| **Model Summary with coded coefficients** | | | | | |
| Term | Coef | SE Coef | P-Value | VIF | R-sq(adj) |
| Constant | 62 | 0.305 | 0 | - | 94.30% |
| SYS_LOSS_VALUE | 9.124 | 0.223 | 0 | 1.13 | - |
| Mass | | | | | |
| 100 kg | 4.751 | 0.444 | 0 | 1.13 | - |
| **Equations** | | | | | |
| 80 kg | ACCEL_COST = 44.203 + 1.2735 SYS_LOSS_VALUE | | | | |
| 100 Kg | ACCEL_COST = 48.954 + 1.2735 SYS_LOSS_VALUE | | | | |

**Table 7. Accel cost: Fixed-wheel turn.**

**Analysis of Variance**

| Source | DF | Adj SS | Adj MS | F-Value | P-Value |
|---|---|---|---|---|---|
| SYS_LOSS_VALUE | 1 | 3481.9 | 3481.87 | 761.65 | 0 |
| WD_CONFIG | 2 | 123.4 | 61.68 | 13.49 | 0 |
| Error | 140 | 640 | 4.57 | - | - |
| Total | 143 | 4238.7 | - | - | - |

**Model Summary with coded coefficients**

| Term | Coef | SE Coef | P-Value | VIF | R-sq(adj) |
|---|---|---|---|---|---|
| Constant | 25.931 | 0.309 | 0 | - | 84.58% |
| SYS_LOSS_VALUE | 4.935 | 0.179 | 0 | 1 | - |
| WD_CONFIG | | | | | |
| 70%DW | -1.611 | 0.436 | 0 | 1.33 | - |
| 80%DW | -2.187 | 0.436 | 0 | 1.33 | - |

**Regression Equations in Uncoded Units**

| WD_CONFIG | |
|---|---|
| 60%DW | ACCEL_COST = 9.69 + 1.383 SYS_LOSS_VALUE |
| 70%DW | ACCEL_COST = 8.08 + 1.383 SYS_LOSS_VALUE |
| 80%DW | ACCEL_COST = 7.50 + 1.383 SYS_LOSS_VALUE |

predictor of acceleration cost, with greater weight distribution on the drive wheels also lowering cost of propulsion during acceleration.

## Differences in cost associated with mass and weight distribution

Basic statistics of the propulsion cost for each maneuver were grouped by system mass and by weight distribution. The propulsion costs were averaged across the different wheel configurations and surfaces. Costs of propulsion varied across mass and weight distribution in a consistent manner across the three canonical maneuvers. For the straight trajectory maneuver, the increase from 80 kg to 100 kg had a medium effect on the main cost and a very large effect on the acceleration cost (**Table 8**). The weight distribution had a small effect size for both acceleration and main costs.

For the fixed-wheel turn maneuver, the effect size of increased mass was large on the main propulsion cost and very large during the acceleration phase (**Table 9**). The effect of weight

**Table 8. Straight trajectory: Cost differences.**

| Variable | AMPS Mass | N | Mean (J/m) | StDev | Effect Size |
|---|---|---|---|---|---|
| MAIN_COST | 80 kg | 72 | 27.0 | 6.16 | - |
| | 100 kg | 72 | 31.7 | 8.06 | 0.65 |
| ACCEL_COST | 80 kg | 72 | 58.9 | 8.09 | - |
| | 100 kg | 72 | 69.8 | 9.79 | 1.21 |
| Variable | WD_CONFIG | N | Mean (J/m) | StDev | Effect Size |
| MAIN_COST | 60%DW | 48 | 30.3 | 7.66 | - |
| | 70%DW | 48 | 29.4 | 7.48 | 0.12 |
| | 80%DW | 48 | 28.4 | 7.46 | 0.25 |
| ACCEL_COST | 60%DW | 48 | 65.2 | 10.83 | - |
| | 70%DW | 48 | 64.4 | 10.55 | 0.07 |
| | 80%DW | 48 | 63.5 | 10.20 | 0.15 |

**Table 9. Fixed-wheel turn: Cost differences.**

| Variable | AMPS Mass | N | Mean (J/rad) | StDev | Effect Size |
|---|---|---|---|---|---|
| MAIN_COST | 80 kg | 72 | 14.8 | 3.41 | - |
| | 100 kg | 72 | 18.1 | 4.60 | 0.79 |
| ACCEL_COST | 80 kg | 72 | 22.1 | 3.75 | - |
| | 100 kg | 72 | 27.2 | 5.70 | 1.06 |
| **Variable** | **WD_CONFIG** | **N** | **Mean (J/rad)** | **StDev** | **Effect Size** |
| MAIN_COST | 60%DW | 48 | 16.7 | 4.66 | - |
| | 70%DW | 48 | 16.3 | 4.39 | 0.08 |
| | 80%DW | 48 | 16.4 | 4.06 | 0.06 |
| ACCEL_COST | 60%DW | 48 | 25.9 | 5.63 | - |
| | 70%DW | 48 | 24.3 | 5.47 | 0.29 |
| | 80%DW | 48 | 23.8 | 5.11 | 0.39 |

distribution was much less, with effect sizes ranging from negligible to small as the weight was shifted closer to the drive wheels.

During the zero-radius turning maneuver, system mass had a very large effect of mass on the total propulsion cost (**Table 10**). Weight distribution had a lesser, but consistent, effect with 70% and 80% weight distributions having lower costs of propulsion. The 70% WD had a medium effect size and the 80% WD had a very large effect size over the 60% WD configuration.

The acceleration phases of the straight and fixed-wheel turn maneuvers show that mass had a very large effect size, consistent with the increase in kinetic energy of the system during this aspect of the maneuver. Weight distribution had a lesser effect during acceleration with small to medium decreases in propulsion cost as the weight distribution increased to 70% and 80%. Zero-radius turning displayed a much more pronounced effect of weight distribution on the total propulsion cost.

## Discussion

This study used three different maneuvers to measure the respective propulsion costs during different types of maneuvering. Each maneuver reflected different types and levels of inertial and energy loss influences. Changes in momentum, embodied by changes in the speed and/or direction of the system mass, has a direct relationship on cost. Though the calculation of propulsion cost for straight (in J/m) and turning (in J/rad) maneuvers cannot be directly compared, the relative amounts of work can be related to their differences in kinetic energies. The straight trajectory required the most overall propulsion effort to perform this maneuver, by a large margin. The straight maneuver involves a pronounced change in translational kinetic energy during the acceleration phase, and the operator must impart work to achieve such a

**Table 10. Zero-radius turns: Cost differences.**

| Variable | AMPS Mass | N | Mean (J/rad) | StDev | Effect Size |
|---|---|---|---|---|---|
| TOTAL_COST | 80 kg | 72 | 9.7 | 1.93 | - |
| | 100 kg | 72 | 12.3 | 2.79 | 1.08 |
| **Variable** | **WD_CONFIG** | **N** | **Mean (J/rad)** | **StDev** | **Effect Size** |
| TOTAL_COST | 60%DW | 48 | 12.5 | 2.70 | - |
| | 70%DW | 48 | 10.8 | 2.41 | 0.67 |
| | 80%DW | 48 | 9.6 | 2.27 | 1.16 |

change in energy. A fixed-wheel turn includes a different type of momentum as the center of mass is accelerated around a fixed radius. This maneuver reflects both translational and rotational kinetic energy and required the second highest level of propulsion cost. The zero-radius turn has a very small rotation radius to minimize translational motion. Instead, the energetic emphasis is on rotational kinetic energy as the wheelchair swivels around the position of the drive wheel axle center. With minimal influence from translational energy, the zero-radius turn maneuver resulted in the lowest propulsion costs.

The large increases in work over distance during the acceleration phases illustrate the effect that changes in kinetic energy have on propulsion cost. Within the straight maneuver acceleration phase, this cost exceeded 60 J/m whereas the cost during the steady-state phase of the maneuver was typically half that, at around 30 J/m. The fixed-wheel turning results show a similar trend, with costs of 22–27 J/rad during the acceleration phase and only 14–18 J/rad in the steady-state phase. As expected, increasing velocity incurs more cost than maintaining motion. When applied to human operation, this highlights the greater force required to initiate a bout of mobility, which makes up an immense portion of daily wheelchair usage–as Sonenblum reports, MWC motion is characterized by frequent starts and stops, and 85% of bouts of mobility travel less than 30 meters [34]. Furthermore, wheelchair users who choose to accelerate quickly will perform more work and apply greater forces to the push-rims. Though this study has no direct relationship to the risk of injury in the upper extremities, some associations can be inferred. For example, users who choose to accelerate rapidly will exert more effort in propulsion, which can increase stresses on the shoulders leading to risk of injury. Likewise, a heaver occupant puts more weight over the wheels and must therefore apply larger forces to the push-rims to overcome the greater energy losses. These greater loads on their upper extremities can potentially lead to higher risk of shoulder injuries [1].

In distinction, energy loss is ever-present and must be overcome during acceleration and steady-state motion for every maneuver. This study assessed cost on two surfaces (tile, carpet), thus it reports the combination of cost over surfaces with different levels of energy loss. All wheelchair maneuvers are impacted by rolling resistance as the wheels roll over the ground. The fixed-wheel and zero-radius turns also add scrub torque as the wheels pivot on the surface during these maneuvers.

The modeling results are consistent in identifying energy loss as the most influential factor in determining energy cost during propulsion. This is an important finding that allows assessment into propulsion cost as it impacts everyday maneuvering. This study introduces a System Loss Value as a means to characterize all the system resistance influences during different maneuvers, so it is reflective of the context of use. All models identified System Loss as a significant predictor and, in the case of straight and fixed-wheel turns, it was the sole predictor of models with very high $R^2$ values. The model of zero-radius turns included our defined System Loss as the most influential predictor but also added the system Yaw Inertia, an inertial parameter. This is consistent with the intended purpose of the maneuver, which reciprocally accelerates and decelerates the wheelchair to reflect the importance of system yaw inertia during a maneuver with a high level of rotational kinetic energy [40].

System Loss values are based upon the frictional losses at the drive wheels and casters. The study measured cost using three drive wheels and four casters which provided variation of energy losses within each mass and weight distribution. Because these values resulted from bench tests of components, the results reflect an opportunity to use component level testing to estimate the system energy loss and propulsion cost of wheelchair systems. This is advantageous because systems level testing of wheelchairs is highly complex and time consuming given the vast number of drive wheels and casters on the market. Indeed, within this study, the use of three maneuvers with ten trials each on two surfaces using a wheelchair configured with

three drive wheels and four casters at two system masses with three weight distributions required a total of 4,320 AMPS trials.

Therefore, a pragmatic approach to estimating the propulsion cost of wheelchairs would involve testing the rolling resistances and scrub torques of casters and drive wheels and calculating the System Loss values for the different trajectory profiles. This calculation would offer a more valid and accurate means to estimate propulsion effort of wheelchairs and be useful to clinicians who must select components when configuring wheelchairs. Using a specific example, let's say a clinician was configuring a wheelchair for a 100 kg client and considering either the 24 x 1-3/8" Standard Pneumatic drive wheel and 5 x 1" Primo caster compared to the 24 x 1" Spinergy with a 5 x 1.5" Primo SR caster. A 70% weight distribution is selected and the client estimates that she spends an equal amount of time on tile and carpeted surfaces. Using the component test results and calculating the System Loss Values, a comparison can be made based upon the strength of the models. For these configurations, the Stock Pneumatic and 5 x 1" Primo combination offers a 14% lower System Loss compared to the Spinergy and 5 x 1.5" Primo SR combination during straight maneuvers, a 5% decrease during fixed-wheel turns, and a 12% decrease during zero-radius turns. The clinician and client can then reflect upon these differences to decide if they are meaningful while assessing other factors in selection such as comfort, impact dampening, aesthetics or cost.

As mass and weight distribution impact the energy losses at the component level, the rolling resistance and scrub torques used in the calculation of System Loss values were based upon measurements taken at different loads that reflected the two masses and three weight distributions. As a result, some assessment of mass and weight distribution is proper when evaluating propulsion cost. Weight distribution is an adjustable configuration on higher end manual wheelchairs. The results clearly indicate that adjusting a chair to place a greater portion of weight on the drive wheels results in a lower propulsion cost during every maneuver. When applying this result to energy loss, it reflects the fact that more loading on the casters, such as in the 60% WD, leads to a greater cost of propulsion. This corroborates findings in a study on energy loss during straight and turning trajectories by Lin *et al.* which documented the influences of weight distribution on system energy loss [21]. A benefit exists in adjusting wheelchairs for greater weight on the drive wheels by shifting the seat further backward [30], moving the rear wheel axle forward [7], or generally bringing the shoulder and drive wheel axle closer to vertical alignment. Lin found that both weight distribution and shoulder position had significant influences on propulsion effort [58]. Both these variables are related to the operator's relationship to the drive wheels. However, increasing the weight on the drive wheels is necessarily accompanied by a loss of pitch stability, resulting in a chair that is more prone to tipping backward. Therefore, this adjustment should only be performed by therapists who are able to address the tradeoff between propulsion effort and safety.

Within the models used in this study, mass and weight distribution were not found to be strong predictors of propulsion cost during the propulsive parts of the maneuvers, despite the distinct modeled occupant masses (80 and 100 kg) and weight distributions (60%, 70%, and 80%). These initial results seem to contradict the findings from Sprigle and Huang that mass and weight distribution influenced the propulsive torques required to accelerate a wheelchair and maintain steady-state velocities, respectively [36]. However, the perceived differences in results actually reflect the means of data analysis. During the main cost analyses, the regression statistics indicated a high covariance between both mass and weight distribution with system resistance. These parameters were therefore omitted from the reported models. This indicates that the impacts of mass and weight distribution on the propulsion cost are present as a function of system resistance. The full exclusion of mass as a predictor variable, however, might

also suggest that the increase in propulsion cost is mostly from the effect of added mass on the resistive properties of the components, rather than from the extra system inertia.

## Conclusion

The work presents statistical models of MWC motion that report the most influential parameters for predicting cost of propulsion. Component-level rolling resistance force and scrub resistive torque tests were performed under realistic loading conditions and combined to represent the frictional losses of the MWC system during three distinct maneuvers. A robotic testing platform collected propulsive cost data over 4,000 trials reflecting MWC configurations using three drive wheels, four casters, two surfaces, two system mass, and three weight distributions, performing three maneuvers. The system loss values, estimated purely from component-level tests, generated linear models strongly correlated ($R^2 > 0.84$) with the costs of propulsion during both acceleration and steady-state phases of motion, and were largely the sole predictor of propulsion cost with the exception of zero-radius turns, which added system yaw inertia. Energy loss is evidently the most influential factor for determining the cost of propulsion of the MWC system. A secondary finding showed that overall system mass had little impact on the cost of propulsion that was not already accounted for by the energy loss parameters, though shifting weight from the casters to the drive wheels resulted in an overall decrease in the cost.

## Supporting information

**S1 Appendix. Canonical AMPS maneuvers.** Descriptions and rationales behind the Straight, Fixed-Wheel Turn, and Alternating Zero-Radius Turns maneuvers.
(PDF)

## Acknowledgments

The authors would like to thank Dr. Teresa Snow for her assistance with the statistical analyses.

## Author Contributions

**Conceptualization:** Morris Huang, Stephen Sprigle.

**Data curation:** Jacob Misch, Morris Huang, Stephen Sprigle.

**Formal analysis:** Jacob Misch, Morris Huang, Stephen Sprigle.

**Funding acquisition:** Stephen Sprigle.

**Investigation:** Jacob Misch, Morris Huang, Stephen Sprigle.

**Methodology:** Jacob Misch, Morris Huang, Stephen Sprigle.

**Project administration:** Stephen Sprigle.

**Resources:** Stephen Sprigle.

**Software:** Jacob Misch.

**Supervision:** Stephen Sprigle.

**Validation:** Morris Huang.

**Visualization:** Jacob Misch, Morris Huang.

**Writing – original draft:** Jacob Misch, Morris Huang, Stephen Sprigle.

**Writing – review & editing:** Jacob Misch, Morris Huang, Stephen Sprigle.

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
