## [Decision Letter · Decision Letter 0]

13 Apr 2020

PONE-D-20-01476

Modeling manual wheelchair propulsion cost during straight and curvilinear trajectories

PLOS ONE

Dear Mr. Misch,

Thank you for submitting your manuscript to PLOS ONE. After careful consideration, we feel that it has merit but does not fully meet PLOS ONE’s publication criteria as it currently stands. Therefore, we invite you to submit a revised version of the manuscript that addresses the points raised during the review process.

We would appreciate receiving your revised manuscript by May 28 2020 11:59PM. To enhance the reproducibility of your results, we recommend that if applicable you deposit your laboratory protocols in protocols.io, where a protocol can be assigned its own identifier (DOI) such that it can be cited independently in the future. For instructions see: http://journals.plos.org/plosone/s/submission-guidelines#loc-laboratory-protocols

We look forward to receiving your revised manuscript.

Kind regards,

Yih-Kuen Jan, PhD

Academic Editor

PLOS ONE

Journal Requirements:

3. We note you have included a table to which you do not refer in the text of your manuscript. Please ensure that you refer to Table 3 to 10 in your text; if accepted, production will need this reference to link the reader to the Table.

Reviewers' comments:

Reviewer's Responses to Questions

**Comments to the Author**

1. Is the manuscript technically sound, and do the data support the conclusions?

Reviewer #1: Yes

Reviewer #2: Yes

2. Has the statistical analysis been performed appropriately and rigorously? 

Reviewer #1: Yes

Reviewer #2: Yes

3. Have the authors made all data underlying the findings in their manuscript fully available?

Reviewer #1: Yes

Reviewer #2: No

4. Is the manuscript presented in an intelligible fashion and written in standard English?

Reviewer #1: Yes

Reviewer #2: Yes

5. Review Comments to the Author

Reviewer #1: Overall, the paper was well written, and the topic is of interest to the research community who studies manual wheelchairs design, wheelchair biomechanics, Propulsion efficiency.

Abstract:

Line 28: Which predictive variables? Name them.

Introduction:

Lines 41-42: does it mean that modifying the tested wheelchair could leads to lack of sensitivity?

Line 75: Is using non-human operators a limitation for the current study as well? If so, it needs to be discussed in the discussion section.

Line 109: “these”? This is a new paragraph, the topic sentence needs to be rephrased.

Lines 136-143: Using a wheelchair propulsion robot should be mentioned in this section.

The necessity/advantage of using wheelchair propulsion robot needs to be justified in the previous paragraphs.

Overarching approach:

Line 187: “diagram above”? Name the figure.

Methods:

Surface quality should be clearly defined. What type of carpet (high pile/low pile), with/without padding …

Line 203: Cite a reference for the instrumented coast-down cart system.

Equation (1): Why the denominator is 4? Are all 4 wheels considered with the same material and contact surface with the ground?

Equation (2): This is not a new equation it is exactly the same as equation (1).

System-level testing of propulsion cost:

A figure or CAD design of the AMPS could help the readers to better understand the system-level testing used in this study. Figures in the supplemental documents are not clear and informative.

Outcome variable:

Line 288: Propulsion cost over distance, time, or both?

Line 293: How was AMPS power calculated / measured?

Steady state and acceleration phase should be clearly defined.

Line 312: Error!

System loss values:

Line 339: Error!

Results:

Line 395: Yaw inertia

Tables 8-10: Are the how was the mean power calculated? Averaged over different wheel and caster conditions?

Discussion:

Line 466: This is not necessarily correct in all ground conditions!

Figure 3: the diagram does not have a quality. Is it just for one stroke cycle (start-up)?

Reviewer #2: General Comment:

The current manuscript is technically challenging and lengthy however the paper is well written, and the authors have done a good job translating their findings. I have only minor suggestions for improved clarity and clinical translation specifically. Propulsion cost was examined through modeling the relationships of inertial and energy-loss parameters across different wheelchair configurations during straight and curvilinear trajectories. More specifically, they examined three wheeling maneuvers over two surfaces using three different drive wheels and four casters types, and two system masses with three weight distributions. Despite the numerous conditions and complex modeling techniques employed, their statistical approach (regression with general linear modeling) was straight forward and pull the data and outcome measures together appropriately (which is much appreciated).

Although the study is a departure from traditional human subjects based propulsion testing, the authors seek to translate their findings clinically. This is where I believe some details could be added to improve the overall clarity and impact of the manuscript. Since a vast majority of the authors findings appear intuitive and support the literature related to the influence of configuration/set up on energy cost and even propulsion technique and upper limb injury, it may be helpful to include more concrete examples of how end users are effected and what clinicians should take away. In particular, the authors could do more to sell their results in a way that readers/clinicians with knowledge of the literature can follow. In its current for the manuscript is heavily engineering intensive, which is fine, however softening some of the example points could improve the readers ability to realize the many practical applications which the data supports.

The following bullet points are examples of impactful results from the current paper that support previous studies related to efficiency, technique, and even upper limb health/pain in MWUs. The authors could elaborate slightly on these points:

• Body weight increases rolling resistance and likelihood of upper limb injury due to increased strain (lines 424-5)

• Acceleration is more straining to upper limbs then steady state propulsion. This was well described by the authors, but they could also stress that in the real world MWU’s propel with more start/stops and accelerations then steady state propulsion which makes their results even more salient (lines 472-473)

• Lines 504-514 is a great example of clinical application, however the authors could briefly mention their findings have implications for upper limb injury prevention as well.

General comment about study goals/aims:

Since the methodology of this study is complex, can the authors present their aims and goals more consistently throughout the paper? Below are examples of how the aims are written slightly differently throughout the paper.

• (abstract)The objective of this study was to model the relationships of inertial and energy-loss parameters to propulsion cost across different wheelchair configurations during straight and curvilinear trajectories

• (lines 136-8) Thus, the goal of this study was to model the mechanical system-level propulsion cost with a relationship between several 'predictor' variables comprised of component- and systems-level parameters

• (lines 158-9) This work describes the use of component- and systems-level parameters as predictors for a statistical model of empirically measured MWC mechanical propulsion cost.

• (lines 552-3) “though shifting weight from the casters to the drive wheels resulted in an overall decrease in the cost” Perhaps add a statement about rear wheel axle position

Lines 269-272- (weight distribution)

Why did the authors select this range of weight distributions relative to the drive wheel? How might this relate to the position of the rear wheel axle relative to the shoulder. Would an 80% load on the drive wheel represent a realistic placement of the rear wheel relative to the person? Would this be a chair that tips backward very easily? Can you simply relate % weight over drive wheel to rear wheel axle position in a way that’s translates more intuitively clinically?

6. PLOS authors have the option to publish the peer review history of their article (what does this mean?). If published, this will include your full peer review and any attached files.

Reviewer #1: Yes: Omid Jahanian

Reviewer #2: No

---

## [Author Response · Author response to Decision Letter 0]

5 May 2020

Note that this content is formatted appropriately in the uploaded "Response to Reviewers.docx" file. 

Journal Requirements:

 Please ensure that your manuscript meets PLOS ONE's style requirements, including those for file naming. The PLOS ONE style templates can be found at http://www.journals.plos.org/plosone/s/file?id=wjVg/PLOSOne_formatting_sample_main_body.pdf and http://www.journals.plos.org/plosone/s/file?id=ba62/PLOSOne_formatting_sample_title_authors_affiliations.pdf

To the best of our knowledge, the manuscript and supporting files meet the style requirements and naming conventions provided by PLOS ONE. 

 We note that you have stated that you will provide repository information for your data at acceptance. Should your manuscript be accepted for publication, we will hold it until you provide the relevant accession numbers or DOIs necessary to access your data. If you wish to make changes to your Data Availability statement, please describe these changes in your cover letter and we will update your Data Availability statement to reflect the information you provide.

The data is prepared for submission to an online repository hosted by Georgia Institute of Technology, which allows free access. The access link will be provided upon acceptance of this manuscript because the link will reference the manuscript citation.

 We note you have included a table to which you do not refer in the text of your manuscript. Please ensure that you refer to Table 3 to 10 in your text; if accepted, production will need this reference to link the reader to the Table.

References to the tables are now included in the preceding paragraphs. 

 Please include captions for your Supporting Information files at the end of your manuscript, and update any in-text citations to match accordingly. Please see our Supporting Information guidelines for more information: http://journals.plos.org/plosone/s/supporting-information.

The supplemental document caption was erroneously placed at the beginning of the supplemental document. It has been relocated to the end of the manuscript. 

 

Comments to the Author

 Is the manuscript technically sound, and do the data support the conclusions?

Reviewer #1: Yes

Reviewer #2: Yes

 Has the statistical analysis been performed appropriately and rigorously? 

Reviewer #1: Yes

Reviewer #2: Yes 

 Have the authors made all data underlying the findings in their manuscript fully available?

The PLOS Data policy requires authors to make all data underlying the findings described in their manuscript fully available without restriction, with rare exception (please refer to the Data Availability Statement in the manuscript PDF file). The data should be provided as part of the manuscript or its supporting information, or deposited to a public repository. For example, in addition to summary statistics, the data points behind means, medians and

variance measures should be available. If there are restrictions on publicly sharing data—e.g. participant privacy or use of data from a third party—those must be specified.

Reviewer #1: Yes

Reviewer #2: No

To respond to Reviewer #2, the data used in the statistical analyses as described in the manuscript will be made available on Georgia Institute of Technology’s digital repository, SMARTech. 

 Is the manuscript presented in an intelligible fashion and written in standard English?

Reviewer #1: Yes

Reviewer #2: Yes 

 Review Comments to the Author

We thank both reviewers for their comments and suggestions on the submitted manuscript. We have made changes to the manuscript to assist with the readability of the material and clarity of the presented information. 

Reviewer #1: 

Overall, the paper was well written, and the topic is of interest to the research community who studies manual wheelchairs design, wheelchair biomechanics, Propulsion efficiency.

Abstract:

Line 28: Which predictive variables? Name them.

This sentence was updated to introduce the predictive variables. The sentences in Lines 29-30 shed more detail on how the variables were used. 

Introduction:

Lines 41-42: does it mean that modifying the tested wheelchair could leads to lack of sensitivity?

This statement was unclear in the original document. Lines 42-44 include two separate statements: first, that some studies such as (5) investigated human propulsion effort but had to heavily modify the MWC beyond its standard configuration, which certainly reduces its applicability towards the field; second, human subjects in general have appeared insensitive to large changes in mass such as an added 10kg to the frame in (6,7). This sentence has been updated to make a clearer distinction between the two claims. 

Line 75: Is using non-human operators a limitation for the current study as well? If so, it needs to be discussed in the discussion section.

We acknowledge the possibility that the use of our non-human operator may appear to be a limitation. However, we have explicitly different goals from most studies that utilize human operators. 

We are focused on the mechanical system of the MWC as a vehicle, separate from the biomechanics of the human operator. In this manuscript, therefore, the AMPS is not a limitation per se, rather is a means to study the mechanical system in a manner that offers a significant advantage over human subject testing. 

The design of the series of experiments used for this manuscript required careful control of targeted variables. In this case, the AMPS provides a repeatable output (in the form of a velocity trajectory) that all tested chairs are subjected to, which would be much less controllable with human operators adapting to different MWC configurations. Additionally, the AMPS can be configured to various system masses or weight distributions to ensure the loading conditions on each wheel is consistent. All of these factors demonstrate why the AMPS is not a limitation, but a necessary apparatus for our experiment. 

That being said, we do need to make a link to clinical relevance. We have included several statements about the impact of configurations, including weight distribution, and caster and drive wheel choices as well as addressing the benefits of not rapidly accelerating when starting to propel from a stop.

Line 109: “these”? This is a new paragraph, the topic sentence needs to be rephrased.

The sentence has been rephrased. 

Lines 136-143: Using a wheelchair propulsion robot should be mentioned in this section. The necessity/advantage of using wheelchair propulsion robot needs to be justified in the previous paragraphs.

A new section has been added to the introduction (Lines 98-102). This gives justification for using the robotic operator. Additionally, a sentence was added (Lines 142-147) that references the AMPS in regards to the overarching goal of this work. 

Overarching approach:

Line 187: “diagram above”? Name the figure.

Figures 1 and 2 are now referenced directly. 

Methods:

Surface quality should be clearly defined. What type of carpet (high pile/low pile), with/without padding …

This is now stated in the “Configurations” section, Line 361. The surface is low-pile carpet laid directly over a polished concrete floor without underlay padding. 

Line 203: Cite a reference for the instrumented coast-down cart system.

The first use of the coast-down cart reference has been moved to the introduction of the coast-down cart in the “Component-level testing: coast-down cart” section. 

Equation (1): Why the denominator is 4? Are all 4 wheels considered with the same material and contact surface with the ground?

Equation 1 describes how we determine the rolling resistance force of each ‘reference’ caster when the cart is equipped with 4 identical caster wheels. In reference (50), it describes how the weight distribution is very carefully controlled so each of the 4 identical wheels are under equivalent loading conditions. 

Equation (2): This is not a new equation it is exactly the same as equation (1).

We acknowledge the similarities between the two equations. Equation 2 is the same basic formula as Equation 1, yes, but they have different purposes. Equation 1 merely gives you the rolling resistance force of a wheel when you have 4 of the exact same wheels equipped. Our cart is not configured for that. We use only one set of ‘reference’ (front) wheels in each and every test. The back wheels can be anything from 4”-diameter casters to 24”-diameter drive wheels. 

The “Component-level testing: coast-down cart” has been revised to elaborate more on the experimental methods. This has improved the context for Equations 1 and 2.

System-level testing of propulsion cost:

A figure or CAD design of the AMPS could help the readers to better understand the system level testing used in this study. Figures in the supplemental documents are not clear and informative.

A figure of the AMPS has been included within the “System-level testing of propulsion cost” section on new Line 266. This figure provides a clear image of the AMPS seated on an ultra-lightweight MWC. The right side of the AMPS is fully visible which shows the motor ‘hand’ meshing with the custom ring gear push-rim. 

Regarding the supplemental documents, the supplemental figures provide more information about the maneuvers and give the readers images of the system-level testing ‘in action’. We believe that these images and descriptions contain useful details for readers who want to learn more about the canonical maneuvers. 

Outcome variable:

Line 288: Propulsion cost over distance, time, or both?

In that same paragraph (new Line 317), the propulsion cost is defined as “…the amount of energy required to perform a maneuver, normalized by the displacement or distance traveled by the center of mass.” The propulsion energy is an integral of the system power over time. Then, this energy value is divided by the distance traveled by the center of mass. 

Line 293: How was AMPS power calculated / measured? Steady state and acceleration phase should be clearly defined.

The AMPS power calculation is described in that paragraph on Line 319: “…the propulsion power is calculated by multiplying the left and right wheel propulsion torques (τ_L, τ_R) with the corresponding angular wheel velocities (ω_L, ω_R)…”

We acknowledge that the acceleration and steady-state phases were not adequately described. They are now defined further down the section in Lines 329-333. 

Line 312: Error!

This comment is unclear. The paragraph had some grammatical/capitalization errors that have been corrected. 

System loss values:

Line 339: Error!

This sentence (Line 371) has been reworded to have a clearer description of what is actually being calculated. 

Results:

Line 395: Yaw inertia

The term “System Yaw” or “System Yaw Inertia” throughout the paper has been properly replaced with Yaw Inertia. 

Tables 8-10: Are the how was the mean power calculated? Averaged over different wheel and caster conditions?

The reviewer’s intuition is correct about the calculations. The mean propulsion cost for 80kg, for example, was found by averaging each trial from the 80kg category, across all 3 weight distributions, 3 drive wheels, 4 casters, and 2 surfaces. The 60% weight distribution averaged across 80kg and 100kg, as well as both surfaces and all the wheel/caster conditions. This section has been revised to include a better explanation of these averages. 

Discussion:

Line 466: This is not necessarily correct in all ground conditions!

This comment is referring to the statement: “With minimal influence from translational energy, the zero-radius turn maneuver resulted in the lowest propulsion costs.” According to the data presented within Tables 8-10, this statement is true in all cases: zero-radius turns had total costs ranging from 9.6-12.5 J/rad, whereas fixed-wheel turns had costs from 14.8-27.2 J/rad, and the straight maneuver had much higher costs from 27.0-69.8 J/m. Though J/rad and J/m are not directly comparable, there doesn’t seem to be enough evidence to refute the quoted statement. 

Figure 3: the diagram does not have a quality. Is it just for one stroke cycle (start-up)?

This is correct. The diagram is showing one push (not a ‘stroke cycle’, because the coast-down cart is not a wheelchair, it is a cart that is given one single push from behind by a human operator that immediately lets go of the cart). 

A more complete description of the coast-down cart is included in this section. In particular, Lines 216-221 elaborate on the methods of collecting one ‘trial’ of coast-down data. The diagram on the right side of Figure 3 shows the velocity of the cart. The rapid acceleration is when the operator is pushing the cart from rest up to the target velocity, at which point the operator releases the cart. The steady downward slope of the velocity represents the relatively constant deceleration of the cart which is used in Equations 1 and 2 as a_cart. 

Reviewer #2: 

General Comment: The current manuscript is technically challenging and lengthy however the paper is well written, and the authors have done a good job translating their findings. I have only minor suggestions for improved clarity and clinical translation specifically. Propulsion cost was examined through modeling the relationships of inertial and energy-loss parameters across different wheelchair configurations during straight and curvilinear trajectories. More specifically, they examined three wheeling maneuvers over two surfaces using three different drive wheels and four casters types, and two system masses with three weight distributions. Despite the numerous conditions and complex modeling techniques employed, their statistical approach (regression with general linear modeling) was straight forward and pull the data and outcome measures together appropriately (which is much appreciated).

Although the study is a departure from traditional human subjects-based propulsion testing, the authors seek to translate their findings clinically. This is where I believe some details could be added to improve the overall clarity and impact of the manuscript. Since a vast majority of the authors findings appear intuitive and support the literature related to the influence of configuration/set up on energy cost and even propulsion technique and upper limb injury, it may be helpful to include more concrete examples of how end users are affected and what clinicians should take away. In particular, the authors could do more to sell their results in a way that readers/clinicians with knowledge of the literature can follow. In its current for the manuscript is heavily engineering intensive, which is fine, however softening some of the example points could improve the readers ability to realize the many practical applications which the data supports.

The authors thank the reviewer for these comments and want to clarify one major point: Our work as presented in this manuscript is focused on studying the MWC as a mechanical system. 

We acknowledge the need for clinical relevance , but we have chosen to include examples specific to work and energy rather than direct inferences about secondary complications. From our propulsion cost metric, we can only make inferences on the biomechanical propulsion ‘effort’ the human user must exert. We have attempted to add examples and clinical relevance within the discussion. 

The following bullet points are examples of impactful results from the current paper that support previous studies related to efficiency, technique, and even upper limb health/pain in MWUs. The authors could elaborate slightly on these points: 

 Body weight increases rolling resistance and likelihood of upper limb injury due to increased strain (lines 424-5)

To address this comment, a section was added to Lines 508-518. It is generally agreed that this edit will benefit the readers by connecting our results to some real-world implications. However, this has been prefaced with the caveat that our results are not directly translatable to upper limb injury. 

 Acceleration is more straining to upper limbs then steady state propulsion. This was well described by the authors, but they could also stress that in the real world MWU’s propel with more start/stops and accelerations then steady state propulsion which makes their results even more salient (lines 472-473)

This is an excellent point and has been added into the section of interest, citing S.E.Sonenblum’s article on how MWC motion features short bouts of mobility.

 Lines 504-514 is a great example of clinical application, however the authors could briefly mention their findings have implications for upper limb injury prevention as well.

The referenced section discusses the differences in system energy loss of two specific configurations under each of the three maneuvers. Because there are no direct links between System Loss and upper limb injury prevention, any statement about injury falls outside of the scope of this work. 

General comment about study goals/aims:

Since the methodology of this study is complex, can the authors present their aims and goals more consistently throughout the paper? Below are examples of how the aims are written slightly differently throughout the paper.

 (abstract) The objective of this study was to model the relationships of inertial and energy loss parameters to the propulsion cost across different wheelchair configurations during straight and curvilinear trajectories

 (lines 136-8) Thus, the goal of this study was to model the mechanical system-level propulsion cost with a relationship between several 'predictor' variables comprised of component- and systems-level parameters

 (lines 158-9) This work describes the use of component- and systems-level parameters as predictors for a statistical model of empirically measured MWC mechanical propulsion cost.

As noted, consistency is key. The descriptions of the objective at each location have been updated to maintain a more consistent language and tone.

(lines 552-3) “though shifting weight from the casters to the drive wheels resulted in an overall decrease in the cost” Perhaps add a statement about rear wheel axle position

In this study, the weight distribution was changed by adjusting where the weight was placed on the AMPS, not by adjusting the rear wheel axle position. This is an important distinction because changing axle position actually impacts multiple parameters, weight distribution and wheelbase, whereas we were able to change only a single parameter. A section was added to Lines 567-574 to cover some changes to the MWC configuration that impact weight distribution. 

Lines 269-272- (weight distribution):

Why did the authors select this range of weight distributions relative to the drive wheel? 

The chosen weight distributions of 60%, 70%, and 80% were based upon clinical relevance and related literature (Brubaker, 1986; Lin & Sprigle, 2020; Sawatzky et al., 2004). 

The lower bound and overall range of the %WD was partially informed by Brubaker, who used a MWC configured at 60%WD over the drive wheels. The same paper works through an example of shifting the center of mass 2.4 inches rearward to adjust to 75%WD, to the effect of a 6% decrease in rolling resistance. Sawatzky’s 2004 article used a 76%WD to simulate a human occupant in the MWC during the experiment. Lin & Sprigle measured the configurations of 23 MWC users and reported a mean of 70% WD.

How might this relate to the position of the rear wheel axle relative to the shoulder? 

Typically, a higher weight distribution percentage over the drive wheels indicates that either the center of gravity is shifted rearward in the chair, or that the rear wheel axle is shifted forward, or both. Therefore, it is logical and correct to assume the position of the drive wheel axle and the weight distribution are highly correlated, as shown in (Lin & Sprigle, 2020)- with both influencing propulsion effort. 

Additional sentences have been added describing this relationship in Lines 567-574. 

Would an 80% load on the drive wheel represent a realistic placement of the rear wheel relative to the person?

In short, yes. An 80%WD is well within realistic loading conditions. Literature on this subject is scarce; the sources used to inform the design of this study show that 70% is about the average WD%, though it is not uncommon for users to have greater weight distributions to achieve greater maneuverability and reduce the energy losses experienced in everyday use. 

Would this be a chair that tips backward very easily? 

Decisions when designing an optimized MWC require compromise – a centralized weight placement of 60%WD puts more weight over the casters and increases the rolling resistance and scrub torques, but is inherently more stable than a chair with 80%WD. In contrast, the 80%WD chair experiences less rolling resistance and scrub torque, is more maneuverable on flat surfaces, and can more easily pop a wheelie to climb curbs. Naturally, this chair will be at greater risk of tipping when ascending an incline. This relationship is stated in the manuscript

Can you simply relate % weight over drive wheel to rear wheel axle position in a way that’s translates more intuitively clinically?

This is a little difficult because the person and how they sit have such a great influence on weight distribution. We have mentioned the general relationship that moving the rear axle forward will shift more weight onto the drive wheels. However the precise amount must be individually determined.

---

## [Decision Letter · Decision Letter 1]

2 Jun 2020

Modeling manual wheelchair propulsion cost during straight and curvilinear trajectories

PONE-D-20-01476R1

Dear Dr. Misch,

We are pleased to inform you that your manuscript has been judged scientifically suitable for publication and will be formally accepted for publication once it complies with all outstanding technical requirements.

With kind regards,

Yih-Kuen Jan, PhD

Academic Editor

PLOS ONE

Additional Editor Comments (optional):

Reviewers' comments:

Reviewer's Responses to Questions

**Comments to the Author**

1. If the authors have adequately addressed your comments raised in a previous round of review and you feel that this manuscript is now acceptable for publication, you may indicate that here to bypass the “Comments to the Author” section, enter your conflict of interest statement in the “Confidential to Editor” section, and submit your "Accept" recommendation.

Reviewer #1: All comments have been addressed

Reviewer #2: All comments have been addressed

2. Is the manuscript technically sound, and do the data support the conclusions?

Reviewer #1: Yes

Reviewer #2: Yes

3. Has the statistical analysis been performed appropriately and rigorously? 

Reviewer #1: Yes

Reviewer #2: Yes

4. Have the authors made all data underlying the findings in their manuscript fully available?

Reviewer #1: Yes

Reviewer #2: Yes

5. Is the manuscript presented in an intelligible fashion and written in standard English?

Reviewer #1: Yes

Reviewer #2: Yes

6. Review Comments to the Author

Reviewer #1: (No Response)

Reviewer #2: The authors have done a good job incorporating reviewer's comments which i believe has improved the manuscripts quality and potential for translation.

7. PLOS authors have the option to publish the peer review history of their article (what does this mean?). If published, this will include your full peer review and any attached files.

Reviewer #1: Yes: Omid Jahanian

Reviewer #2: Yes: Ian Matthew Rice

---

## [Editor Report · Acceptance letter]

8 Jun 2020

PONE-D-20-01476R1 

Modeling manual wheelchair propulsion cost during straight and curvilinear trajectories 

Dear Dr. Misch:

I'm pleased to inform you that your manuscript has been deemed suitable for publication in PLOS ONE. Congratulations! Your manuscript is now with our production department. 

Kind regards, 

on behalf of

Dr. Yih-Kuen Jan 

Academic Editor

PLOS ONE